# Risk factors associated with the discordance in kidney function decline rate in identical twins

Jeong Ah Hwang[1]ⓘ, Jaeun Shin[1]ⓘ, Eunjung Cho[1], Shin Young Ahn[1,2], Gang-Jee Ko[1,2], Young Joo Kwon[1,2], Ji Eun Kim ⓘ[1,2] *

1 Department of Internal Medicine, Korea University Guro Hospital, Seoul, Republic of Korea, 2 Department of Internal Medicine, Korea University College of Medicine, Seoul, Republic of Korea

ⓘ These authors contributed equally to this work.
* beeswaxag@naver.com

## Abstract

### Background

The rate of kidney function decline is different for each individual regardless of any difference in the medical histories. This study set out to identify the risk factors for high discordance in kidney function decline in an identical twin cohort.

### Methods

This study included 333 identical twins from the Korean Genome and Epidemiology Study who were categorized into two groups according to the estimated glomerular filtration rate (eGFR) decline: the slow and rapid progressor groups. The mean differences of variables were compared between the two groups. We calculated the difference in the annual eGFR change between twins and analyzed the risk factors associated with high discordance in twins who had > 5 mL/min/1.73 $m^2$ /yr of the intra-twin difference in the annual eGFR decline. Identical twins with diabetes and baseline eGFR < 60 mL/min/1.73 $m^2$ were excluded.

### Results

The high discordance twins showed significant differences in body mass index; waist-to-hip ratio; total body fat percentage; and levels of blood hemoglobin, serum fasting glucose, albumin, triglyceride, and uric acid; however, there were no differences in low discordance twins. Multivariable logistic regression showed that blood hemoglobin level is the only significant factor associated with high discordance of eGFR decline in twins.

### Conclusions

Blood hemoglobin level may play a role in the individual differences in kidney function decline.

**Data Availability Statement:** The data supporting the findings of this study are available in clinical database from Korean Ge-nome and Epidemiology Study and are accessible after permission from http://is.kdca.go.kr (accessed on Sep 2022).

**Funding:** This study was supported by a grant from the KOREAN NEPHROLOGY RESEARCH FOUNDATION (Young Investigator Research Grant, 2022). The funders played no role in the conduct of the study, and the study was performed independently by the authors.

**Competing interests:** The authors have declared that no competing interests exist.

## Introduction

Chronic kidney disease (CKD), which is defined as a kidney dysfunction of over 3 months, is an important public health issue [1, 2]. CKD is a complex disease with heterogeneous and ambiguous clinical presentations. There are various clinical diseases related to the development of CKD, such as diabetes, hypertension, and ischemic cardiovascular diseases, however, some patients with CKD have an elusive etiology [3].

CKD is quite common in the general population worldwide, with an estimated global prevalence of 11% to 13% [2]. The annual estimated glomerular filtration rate (eGFR) decline has been heterogeneously reported in previous studies but generally declines at a rate of 1 mL/min/year [4]. However, the rate of kidney function decline is different per individual, and the reason for the difference remains unclear [5].

Recently, novel methods and tools for genetic testing have evolved, and the effort to find the genetic etiology in CKD using these tools is increasing [6]. Nevertheless, the mendelian etiology of kidney disease and the novel findings in genetic mutations related to kidney dysfunction only account for a part of CKD cases [3, 7–10], and a large proportion of CKD etiologies remain unelucidated.

In this study, we aimed to evaluate the possible non-genetic factors associated with annual kidney function decline in the general population besides natural aging using an identical twin cohort.

## Materials and methods

### Study setting and study cohort

This observational analysis included a prospective cohort of twins from the Korean Genome and Epidemiology Study (KoGES). The KoGES twin and family cohort consisted of 3,399 individuals who were twins or other family members related to twins. In this cohort, health screenings and surveys were conducted at baseline (2005–2013) and follow-up (2008–2014). The general characteristics of the KoGES twin and family cohort have been published elsewhere [11].

The inclusion criteria were monozygotic twins with both baseline and follow-up laboratory measurement data. The exclusion criteria were fraternal twins, one or both of the twins having diabetes or baseline kidney dysfunction and those who had error in datasets. The definition of kidney dysfunction in exclusion criteria was eGFR $< 60\text{mL/min/1.73m}^2$, calculated using the Chronic Kidney Disease Epidemiology Collaboration (CKD-EPI) equation at baseline [12]. Diabetes is a major factor associated with kidney dysfunction; thus, we excluded the participants who had diabetes. Diabetes was defined as a self-report of diabetes or baseline serum fasting glucose of $\geq 126$ mg/dL. The flow diagrams for the inclusion and exclusion criteria are shown in Fig 1.

This study was exempted from a review from the institutional review board of Korea University Guro Hospital (IRB No. 2021GR0532) and informed consent was waived due to the use of public cohort data that could not identify the participants.

### Data collection

Demographic and social characteristics were retrieved, such as age, sex, body mass index (BMI), waist-to-hip ratio (WHR), alcohol consumption status, smoking status, and medical histories, including hypertension, dyslipidemia, cerebral infarction, and myocardial infarction. Baseline serum and urine laboratory tests were retrieved, including hemoglobin, glucose, albumin, total cholesterol, LDL cholesterol, triglyceride, and uric acids. Daily nutrient intake was calculated and processed using a food frequency questionnaire. The validity and

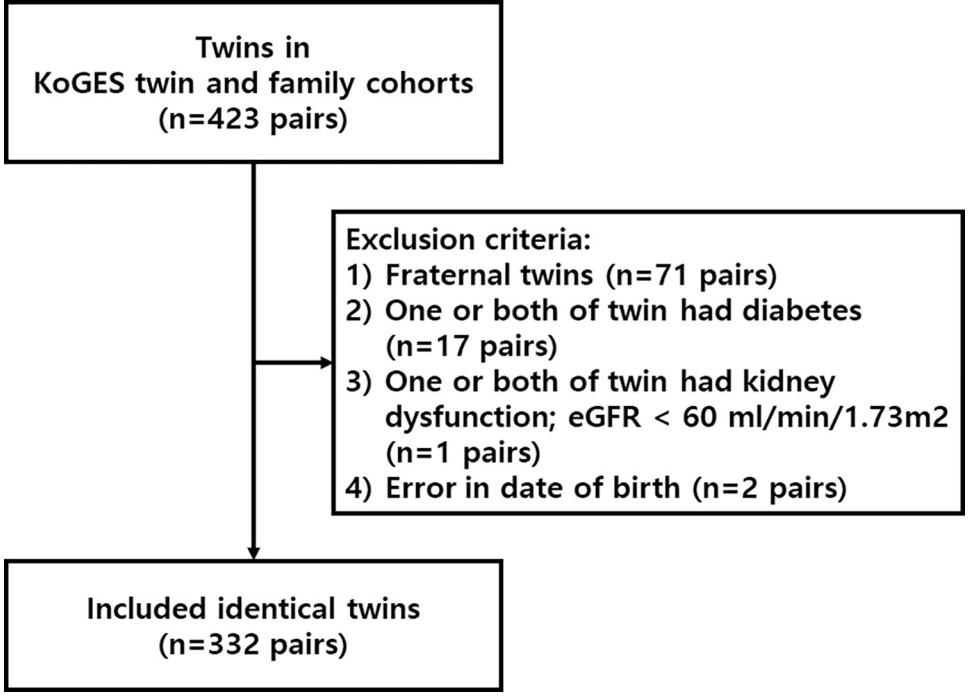

**Fig 1. Flow diagram for participant selection.**

reproducibility of this food frequency questionnaire are provided in a previous study [13]. Each nutrient intake was normalized by dividing the participant's raw data by body weight.

The annual eGFR decline rate was calculated using the formula: (eGFR at follow-up—eGFR at baseline)/follow-up duration (year). The intra-twin difference in eGFR decline rate was calculated by subtracting eGFR decline rate of the slow progressor twin from the eGFR decline rate of the other twin (who is the rapid progressor). Intra-twin difference of other baseline clinical characteristics were also calculated using the formula: (value in 'Slow progressor'—value in 'Rapid progressor').

## Study group classification

All twins were assigned to either the rapid or slow eGFR change groups based on their annual eGFR decline rate. For example, when twins have an eGFR decline rate of 1 and 3 mL/min/1.73 m$^2$/yr, then one of the twins with an eGFR decline rate of 3 mL/min/1.73 m$^2$/yr is included in the rapid eGFR change group and the other twin with an eGFR decline rate of 1 mL/min/1.73 m$^2$/yr was included in the slow eGFR change group. For pair of twins with the same annual eGFR decline rate, we randomly included each twin in each group, rather than manually selecting those with slow or rapid eGFR progression based on the aforementioned criteria.

After classifying rapid or slow eGFR changes in all twins, we defined high-discordant twins as the intra-twin difference of eGFR decline rate > 5 mL/min/1.73 m$^2$/yr. Likewise, low-discordant twins were defined as twins with an intra-twin difference in eGFR reduction of 5 mL/min/1.73 m$^2$/yr or less. These classification criteria for twins with high or low discordance are based on the definition of rapid progression in the 2012 KDIGO Chronic Kidney Disease Guidelines [14]. These differences in eGFR decline rate within twins were classified regardless of their respective absolute eGFR decline rate.

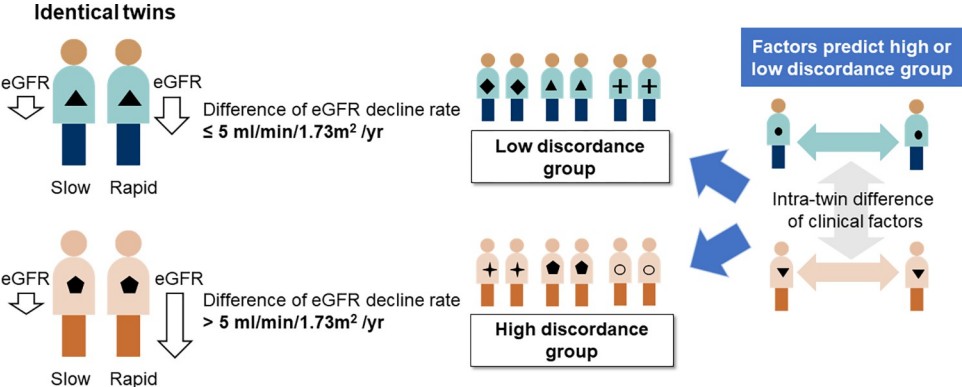

**Fig 2. Schematic flow for study group classification and analysis.**

The method for the study group classification is briefly outlined in Fig 2.

## Study outcome

This study aimed to determine the non-genetic risk factor which predicts the rapid progression of kidney function. Therefore, the main outcome of this study is the risk for high discordance twins who have a disparity of eGFR decline rate of >5 mL/min/1.73m2 /yr. The association between outcome (high discordance group) and various clinical and demographic characteristics was assessed."

## Statistical analysis

Continuous variables are shown as means with standard deviations or medians with interquartile ranges. Categorical variables are presented as numbers and percentages. The missing rates of variables were 0–3.0%. The method used for handling missing data was complete case analysis, disregarding cases with partially missing data. A pairwise comparison between twins was performed using the paired t-test. Univariate and multivariable logistic regression analyses were conducted to find risk factors for high discordance twins. Age, sex, and significant variables ($p < 0.1$) in the univariate analysis were adjusted for multivariable analysis. For logistic regression analyses, all baseline characteristics except for age and sex were included as covariates in the form of intra-twin differences [(value in slow progressor)-(value in rapid progressor)]. Statistical significance was set at $p < 0.05$. All statistical analyses were performed using Stata version 15.0 (StataCorp LLC, TX, US), and the graph was drawn using GraphPad Prism version 9.0.0 (GraphPad Software, Inc., CA, US).

## Results

### Baseline characteristics and annual eGFR change in twins

This study included 332 monozygotic twins (male, 107 twins; mean age, 38.1±7.3 years) in the KoGES twin and family cohort after applying the inclusion and exclusion criteria. The mean annual eGFR change in the slow and rapid progressor groups were -0.62 [-2.5–1.28] mL/min/1.73 m$^2$ and -2.79 [-5.54 –-0.88] mL/min/1.73 m$^2$, respectively. The minimum and maximum intra-twin differences in annual eGFR change were 0 mL/min/1.73 m$^2$ and 33.0 mL/min/1.73 m$^2$, respectively. A total of 8 pairs (2.4%) of twins showed exactly the same eGFR change rates (0 mL/min/1.73 m$^2$ in the intra-twin difference of annual eGFR change).

### Baseline characteristics of the low and high discordance twins stratified by the rate of annual kidney function decline as either a slow progressor or rapid progressor

There were 51 twins (15.4%) in the high discordance group. The baseline characteristics of twins in the high and low discordance groups were listed in Table 1. In the low discordance group, all baseline characteristics, including comorbidities, laboratory findings, and nutritional consumption status, showed no intra-twin differences except baseline eGFR. Meanwhile, in the high discordance group, the slow progressor participants showed a higher BMI, waist-hip ratio, body fat percentage, lower eGFR, and higher levels of blood hemoglobin, serum fasting glucose, albumin, total cholesterol, triglyceride, and uric acid than those in the rapid progressor group. Hypertension was more frequent in the slow progressor group than in the rapid progressor group, while systolic and diastolic blood pressures were not different in both groups.

### Risk analysis for high discordance in eGFR decline in twins

In univariable logistic regression analysis, the high intra-twin differences in BMI, total body fat, HTN and levels of baseline eGFR, hemoglobin, fasting glucose, triglyceride, and uric acid were significantly associated with high discordance of eGFR decline. In multivariable logistic regression, a higher difference in baseline eGFR, blood fasting glucose, and hemoglobin levels was the significant factor associated with the high discordance group. For every 1 ml/min/1.73 $m^2$ increase in baseline eGFR, the risk of the high discordance group was reduced by 11%. For every 1 mg/dL increase in blood glucose difference (higher glucose level in the slow progressor group than in the rapid progressor group), the risk for the high discordance in eGFR decline increased by 1.07 times (p = 0.013). In a similar way, the increased difference of hemoglobin level by 1 g/dL was associated with 1.45 times risk for high discordance group (p = 0.027) (Table 2). Fig 3 shows intra-twin differences in blood glucose and blood hemoglobin levels in the high and low discordance groups.

## Discussion

This prospective Korean cohort study on twins provides intra-twin differences in annual eGFR changes. There were several possible risk factors associated with the high discordance of eGFR decline (> 5 mL/min/1.73 $m^2$ /yr) in twins, including BMI, WHR, total body fat, and levels of hemoglobin, triglyceride, and uric acid. Low blood glucose and hemoglobin levels in the rapid progressor group were significantly associated with the increased risk for high intra-twin eGFR decline rate discordance compared to the slow progressor group even after multivariable adjustment.

Monozygotic twins arise from the same single cell and share most of their genetic variants. However, monozygotic twins show phenotypic discordance for many traits from birth weight to complex diseases [15]. This discordance in an isogenic individual can be caused by somatic mutation, which shows different genotypes by different organs [15, 16] and epigenetic changes, such as DNA methylation and multiple environmental factors [17]. The classic approach to define the contribution of genetic and environmental factors to complex human diseases is to compare the clinical phenotypes in monozygotic twins. Because twins have the most similar genetic factors related to diseases, although epigenetic factors cannot be ignored, the present study with a twin cohort may represent possible environmental factors affecting the phenotype of kidney function decline without previously known renal insufficiency or diabetes.

**Table 1. Comparison of baseline characteristics between slow and rapid eGFR progressor in twins stratified by high or low discordance of eGFR decline rate.**

| | Low discordance twins (Intra-twin difference of annual eGFR decline < 5%) | | | High discordance twins (Intra-twin difference of annual eGFR decline ≥ 5%) | | |
|---|---|---|---|---|---|---|
| | Slow progressor n = 281 | Rapid progressor n = 281 | Paired t-test p-value | Slow progressor n = 51 | Rapid progressor n = 51 | Paired t-test p-value |
| Annual eGFR change, % | -0.7 [-2.5–0.7] | -2.2 [-4.5 - -0.7] | <0.001 | 1.2 [-2.4–4.6] | -7.6 [-9.1 - -2.9] | <0.001 |
| **Demographics and Baseline Clinical Characteristics** | | | | | | |
| Age, year | 38.1±7.3 | 38.1±7.3 | | 37.8±7.4 | 37.8±7.4 | |
| Male sex, n (%) | 94 (33.5) | 94 (33.5) | | 13 (25.5) | 13 (25.5) | |
| Body mass index, kg/m$^2$ | 22.9±3.0 | 22.9±3.1 | 0.717 | 23.3±2.9 | 22.5±2.4 | 0.013 |
| Waist-to-hip ratio, cm/cm | 0.83±0.06 | 0.83±0.08 | 0.866 | 0.83±0.06 | 0.83±0.06 | 0.261 |
| Body fat percentage, % | 28.1±7.5 | 28.4±7.6 | 0.238 | 30.8±7.9 | 29.3±9.0 | 0.111 |
| Smoking status, n (%) | | | 0.665 | | | 0.811 |
| Never | 189 (67.3) | 190 (67.9) | | 38 (74.5) | 40 (78.4) | |
| Ex | 30 (10.7) | 33 (11.8) | | 6 (11.8) | 3 (5.9) | |
| Current | 62 (22.1) | 57 (20.4) | | 7 (13.7) | 8 (15.7) | |
| Alcohol consumption, n (%) | | | 0.103 | | | 0.159 |
| Never | 65 (23.2) | 81 (29.0) | | 14 (27.5) | 8 (15.7) | |
| Ex | 24 (8.6) | 16 (5.7) | | 3 (5.9) | 5 (9.8) | |
| Current | 191 (68.2) | 182 (65.2) | | 34 (66.7) | 38 (74.5) | |
| Systolic blood pressure, mmHg | 111.0±13.8 | 111.2±14.5 | 0.917 | 113.5±13.4 | 112.9±12.3 | 0.708 |
| Diastolic blood pressure, mmHg | 71.3±10.2 | 71.1±10.1 | 0.585 | 70.5±9.1 | 70.8±9.2 | 0.826 |
| **Clinical Histories** | | | | | | |
| Myocardial infarction, n (%) | 0 (0) | 3 (1.1) | 0.083 | 1 (2.0) | 0 (0) | 0.322 |
| cerebral infarction, n (%) | 0 (0) | 1 (0.4) | 0.318 | 1 (2.0) | 0 (0) | 0.322 |
| Hypertension, n (%) | 13 (4.6) | 17 (6.1) | 0.347 | 7 (13.7) | 3 (5.9) | 0.044 |
| Dyslipidemia, n (%) | 11 (3.9) | 19 (6.8) | 0.099 | 3 (5.9) | 0 (0) | 0.083 |
| **Laboratory Findings** | | | | | | |
| Baseline eGFR, mL/min/1.73 m$^2$ | 93.4±13.1 | 96.2±12.8 | <0.001 | 92.9±14.1 | 103.3±12.3 | <0.001 |
| Hemoglobin, g/dL | 13.9±1.6 | 13.8±1.7 | 0.505 | 13.8±1.9 | 13.3±1.9 | 0.004 |
| Glucose, mg/dL | 88 [84–93] | 88 [83–93] | 0.605 | 88 [84–95] | 86 [82–90] | <0.001 |
| Albumin, g/dL | 4.6±0.3 | 4.6±0.3 | 0.1 | 4.6±0.2 | 4.6±0.3 | 0.02 |
| Total cholesterol, mg/dL | 185.5±36.4 | 184.6±36.8 | 0.624 | 192.3±32.7 | 183.4±32.0 | 0.016 |
| LDL cholesterol, mg/dL | 114.1±34.8 | 113.5±33.1 | 0.71 | 114.2±32.3 | 112.3±29.6 | 0.546 |
| Triglyceride, mg/dL | 100.5±73.5 | 97.8±56.3 | 0.556 | 118.0±94.0 | 83.4±52.7 | 0.007 |
| Uric acid, mg/dL | 4.7±1.3 | 4.7±1.4 | 0.981 | 4.8±1.7 | 4.3±1.7 | <0.001 |
| **Nutrition intake (body weight adjusted)** | | | | | | |
| Total calorie intake, kcal | 32.8±14.5 | 33.5±15.2 | 0.723 | 32.2±11.8 | 31.9±14.7 | 0.719 |
| Protein, g | 1.1±0.7 | 1.2±0.6 | 0.664 | 1.1±0.5 | 1.2±0.7 | 0.787 |
| Fat, g | 0.6±0.5 | 0.6±0.4 | 0.666 | 0.6±0.4 | 0.6±0.4 | 0.773 |
| Sugar, g | 5.6±2.3 | 5.7±2.6 | 0.796 | 5.5±2.0 | 5.4±2.3 | 0.646 |
| Ca, mg | 8.9±8.0 | 8.7±7.6 | 0.812 | 7.7±3.9 | 9.0±6.8 | 0.254 |
| P, mg | 17.1±10.4 | 17.5±10.0 | 0.715 | 16.3±6.6 | 17.3±10.3 | 0.711 |
| Fe, mg | 0.2±0.2 | 0.2±0.1 | 0.986 | 0.2±0.1 | 0.2±0.1 | 0.613 |
| K, mg | 44.3±34.1 | 46.0±30.8 | 0.551 | 40.0±19.1 | 45.8±30.7 | 0.303 |
| Na, mg | 48.7±29.1 | 50.1±32.6 | 0.529 | 44.5±23.0 | 48.5±32.8 | 0.576 |

Abbreviations: eGFR, estimated glomerular filtration rate; LDL, low density lipoprotein.

**Table 2. Logistic regressions for high discordance twins in eGFR decline rate according to intra-twin difference in each clinical variable.**

| | Univariable logistic | | Multivariable logistic | |
|---|---|---|---|---|
| | OR (95% CI) | p-value | OR (95% CI) | p-value |
| **Age** | 0.99 (0.95–1.04) | 0.774 | 0.99 (0.94–1.03) | 0.573 |
| **Sex** | 1.47 (0.75–2.89) | 0.41 | 1.67 (0.74–3.76) | 0.216 |
| **Intra-twin difference of values (value in 'Slow progressor'—value in 'Rapid progressor')** | | | | |
| **Anthropometric factors** | | | | |
| Waist-hip ratio | 9.27 (0.03–2593.89) | 0.439 | | |
| Body mass index | 1.22 (1.05–1.42) | 0.008 | 1.02 (0.83–1.25) | 0.886 |
| Body fat percentage | 1.08 (1.01–1.15) | 0.019 | 1.00 (0.93–1.08) | 0.937 |
| Systolic blood pressure | 1.00 (0.98–1.03) | 0.71 | | |
| Diastolic blood pressure | 0.99 (0.96–1.03) | 0.683 | | |
| **Hypertension** | 3.91 (1.26–12.15) | 0.018 | 1.97 (0.50–7.77) | 0.335 |
| **Laboratory findings** | | | | |
| eGFR | 0.89 (0.85–0.93) | <0.001 | 0.89 (0.85–0.94) | <0.001 |
| Hemoglobin | 1.46 (1.11–1.91) | 0.006 | 1.45 (1.04–2.03) | 0.027 |
| Albumin | 2.56 (0.79–8.30) | 0.118 | | |
| Total cholesterol | 1.01 (1.00–1.02) | 0.082 | | |
| Glucose | 1.09 (1.05–1.14) | <0.001 | 1.07 (1.01–1.13) | 0.013 |
| Triglyceride | 1.01 (1.00–1.01) | 0.008 | 1.00 (1.00–1.01) | 0.14 |
| LDL cholesterol | 1.00 (0.99–1.01) | 0.761 | | |
| Uric acid | 2.18 (1.47–3.25) | <0.001 | 1.07 (0.65–1.78) | 0.783 |
| **Nutritional intake per body weight** | | | | |
| Total calorie intake | 1.00 (0.99–1.02) | 0.667 | | |
| Protein | 0.99 (0.70–1.42) | 0.977 | | |
| Fat | 1.10 (0.69–1.77) | 0.689 | | |
| Sugar | 1.02 (0.92–1.14) | 0.642 | | |
| Ca | 0.99 (0.95–1.02) | 0.409 | | |
| P | 1.00 (0.97–1.02) | 0.898 | | |
| Fe | 0.74 (0.13–4.40) | 0.743 | | |
| K | 1.00 (0.99–1.01) | 0.669 | | |
| Na | 1.00 (0.99–1.01) | 0.877 | | |

* Adjusted for age, sex and variables which shown p<0.01 in univariable logistic analysis.

Abbreviations: OR, odds ratio; CI, confidence interval; LDL, low density lipoprotein.

In comparing baseline characteristics between groups, we found that several factors related to malnutrition, such as BMI, waist-hip ratio, body fat percentage, hemoglobin, fasting glucose, and triglyceride, showed significant differences in the high discordance group. In previous studies, body composition metrics, such as BMI, WHR, and total body fat percentage, were significantly correlated with individual nutritional status. In previous studies, body composition metrics, such as BMI, WHR, and total body fat percentage, were significantly correlated with individual nutritional status [18, 19]. Moreover, these metrics seem to be important mediators for the eGFR decline rate in the present study, whose results are consistent with those of previous studies. Several studies have shown that any gain in body weight is associated with better survival in CKD, and that fat-free lean body mass is essentially representative of muscle mass and confers a survival advantage [20]. In a national cohort of US veterans with an eGFR of >60 mL/min/1.73 m$^2$, the lowest risk for loss of kidney function was noted in patients with BMI levels between 25 and 30 kg/m$^2$, whereas a consistent U-shaped association between

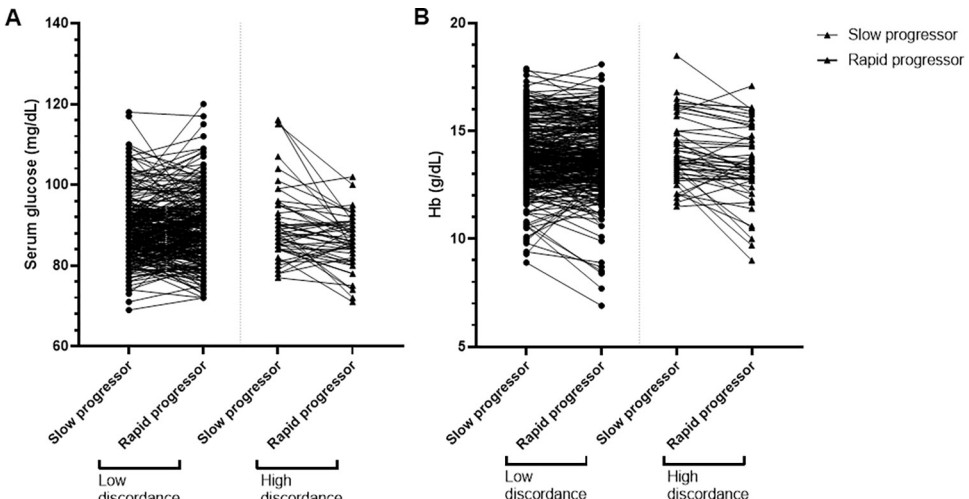

**Fig 3. Blood glucose and hemoglobin levels according to the difference in eGFR decline in twins.** With the dotted line as the center, the plot on the left and right represent (A) glucose and (B) hemoglobin levels in the low and high discordance groups, respectively. Circular dots indicate the slow progressor among the twins, whereas the triangular dots indicate the rapid progressor. The lines connecting the circle and triangle dots represent each pair of twins.

BMI and a rapid loss of kidney function was noted for BMI levels of $<25$ and $>30$ kg/m$^2$ [20]. Additionally, a U-shaped association between BMI and kidney function was noted in another US veteran cohort [21]. A study with Korean nationwide population showed a higher risk for end-stage renal disease in underweight patients with diabetes [22]. Low blood hemoglobin, fasting glucose, triglyceride, and uric acid levels also suggest nutritional impairment according to previous studies [23–26]. Although twins report similar daily nutritional consumption status, nutritional markers found in their blood were significantly decreased, especially in individuals included in the rapid progressor group. This finding suggests that differences in intestinal absorption status between the twins may affect the rate of kidney function decline. Moreover, several studies on the gut-kidney axis associated with gut microbiota affecting nutrient absorption have been reported [27–29]. Further evaluation of the interplay between kidney function and nutritional assessment and intestinal absorption status is warranted.

In addition, we found that blood glucose and hemoglobin levels were associated with eGFR decline rates even after adjusting for several nutritional markers and body composition metrics. Low blood glucose is known to be associated with increased mortality, CVD and ESRD in diabetes patients as well as the increased risk for the development of new-onset diabetes [30]. In a large nationwide T2DM cohort study, clinically significant hypoglycemia was associated with a 1.8 and 1.5-fold greater risk of developing incident CKD and all-cause mortality [31, 32], and post hoc analysis of Action in Diabetes and Vascular disease: PreterAx and DiamicroN Controlled Evaluation (ADVANCE) study showed that hypoglycemia was associated with increased risks of micro- and macro-vascular events [33]. However, hypoglycemia has not been well studied in the general population without diabetes. Even after patients with diabetes were excluded from this study cohort, low blood glucose was associated with a rapid eGFR decline. Although, the exact mechanism between low blood glucose and renal dysfunction is not well understood, some potential relationships may be proposed based on previous studies [34–38]. For example, a preclinical study has demonstrated that hypoglycemia can cause elevated nonesterified fatty acid in adipose tissue, which is subsequently associated with kidney damage [34]. Hypoglycemia could induce sympathetic surges, altering renal hemodynamics, which may be responsible for the progression of kidney dysfunction [35].

Furthermore, low blood glucose might indicate subclinical damage to kidney. The role of the kidney is critical to maintaining glucose homeostasis. The kidney has gluconeogenesis capability and contributes to 20% glycogen production [36]. In the proximal segment of the tubular nephron, SGLT1 and SGLT2 provide highly efficient glucose reabsorption [36–38]. In this context, hypoglycemia may be an early sign of kidney dysfunction. Anemia is a well-known risk factor for the progression of CKD. A prospective observational case-control study reported a greater annual decline in calculated creatinine clearance in patients with diabetes and a hemoglobin level of $< 12$ g/dL than in those with a hemoglobin level of $\geq 12$ g/dL [39]. A randomized clinical trial for early and deferred treatment of anemia in CKD showed that early treatment may delay the progression of kidney function decline [40]. However, there is only limited number of studies on the association between anemia and kidney function decline in the general population without prior kidney disease. Recently, one Chinese population study showed that anemia and hemoglobin were independently associated with the rapid decline in kidney function after adjusting for potential confounding factors [41]. The mechanisms underlying the association between anemia and CKD progression are unclear. However, it has been hypothesized that anemia causes tissue hypoxia and consequently leads to alterations in gene expression patterns, triggering adaptive pathways related to hypoxia-inducing factors in parallel with the induction of noxious mediators involved in the progression of renal injury. The present study not only supports the previous findings but highlights the role of hypoglycemia and anemia as an environmental phenotype rather than a genetic variation affecting kidney function in the general population.

Although it remains unclear which pathophysiologic mechanism of kidney function is affected by anemia, some hypotheses might be suggested. Kidney is somehow sensitive to changes in oxygen delivery [42]. Anemia impairs oxygen delivery to tissues and thus affects organ function, including cardiac function. A triad of worsening anemia, CKD, and cardiac function induces a vicious cycle referred to as the cardiorenal anemia syndrome [43]. Anemia in kidney disease may accelerate the decline in renal function by inducing tubulointerstitial hypoxia [42]. Additionally, anemia may be associated with chronic inflammation [44]; although we do not know the clear reason for this association, we hypothesized that hemoglobin is an early detector of a rapid decline in eGFR and should be considered important when managing the nutritional and environmental status of individuals to maintain kidney function.

To the best of our knowledge, this is the first study to find the environmental factors associated with kidney function decline in the general population using a twin cohort. However, this study has several limitations. First, the genetic information of the twins was not available. As we previously mentioned, somatic mutations or epigenetic differences in twins might have affected the results of this study. Second, the reason for the differences in the hemoglobin levels could not be ascertained due to the lack of data, such as iron profiles, folic acid, vitamin B12, and peripheral blood smears. Though we assessed the food consumption data in this cohort, other nutritional factors, including gut microbiota and their metabolic function which may be related to the environmental difference in twins, need further evaluation." Third, the cause of kidney dysfunction is not known because the cohort only has two time- point measurements (baseline and follow up) and lacks urinalysis. Furthermore, some of the comorbidities related to kidney dysfunction, such as heart failure and glomerulonephritis, were unavailable. Finally, although our study evaluated a prospective cohort with a follow-up dataset, the causal relationship between hemoglobin levels and kidney function decline cannot be suggested due to the retrospective study design.

## Conclusions

In this identical twin cohort with little genetic difference, some of the twins showed high discordance of progression rate of kidney dysfunction. Environmental and nutritional differences affecting fasting glucose and hemoglobin levels may be important clues for this discordance between twins. Therefore, further analysis of the pathophysiology of glucose and hemoglobin metabolism in terms of environmental changes affecting kidney damage is warranted.

## Author Contributions

**Conceptualization:** Jeong Ah Hwang, Ji Eun Kim.

**Data curation:** Jeong Ah Hwang, Jaeun Shin, Ji Eun Kim.

**Formal analysis:** Jaeun Shin.

**Investigation:** Jeong Ah Hwang, Eunjung Cho, Ji Eun Kim.

**Writing – original draft:** Jeong Ah Hwang, Ji Eun Kim.

**Writing – review & editing:** Eunjung Cho, Shin Young Ahn, Gang-Jee Ko, Young Joo Kwon.

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
