## [Decision Letter · Decision Letter 0]

4 Jul 2022

PONE-D-22-15736Risk factors associated with the discordance in kidney function decline rate in identical twinsPLOS ONE

Dear Dr. Kim

Thank you for submitting your manuscript to PLOS ONE. After careful consideration, we feel that it has merit but does not fully meet PLOS ONE’s publication criteria as it currently stands. Therefore, we invite you to submit a revised version of the manuscript that addresses the points raised during the review process. Your submission has been reviewed by two external peer reviewers and myself. The reviewers raises a number of very detailed points particularly concerning methodology and statistical analysis, some of which may require considerable work in returning to the data, if the requested data even exists:State how participants with chronic kidney disease, heart failure and glomerulonephritis were excludedState how acute deterioration of kidney function in study participants was excludedDefine progression according to 2012 KDIGO Clinical Practice GuidelineState the baseline eGFR for the cohortProvide data on urinary protein excretion rateState how sample size/power was calculatedRe-analysis of multivariate logistic regression model based on the comments provided by the reviewersPlease submit your revised manuscript by Aug 18 2022 11:59PM. If you will need more time than this to complete your revisions, please reply to this message or contact the journal office at plosone@plos.org. Please include the following items when submitting your revised manuscript:A rebuttal letter that responds to each point raised by the academic editor and reviewer(s). You should upload this letter as a separate file labeled 'Response to Reviewers'.A marked-up copy of your manuscript that highlights changes made to the original version. You should upload this as a separate file labeled 'Revised Manuscript with Track Changes'.An unmarked version of your revised paper without tracked changes. You should upload this as a separate file labeled 'Manuscript'.

We look forward to receiving your revised manuscript.

Kind regards,

Muzamil Olamide Hassan

Academic Editor

PLOS ONE

Journal Requirements:

Additional Editor Comments:

Ji Eun Kim and colleagues conducted a retrospective identical twins cohort study to assess the risk factors associated with the discordance in kidney function decline rate in identical twins. They found that blood hemoglobin level may play a role in the individual differences in kidney function decline. However, the authors should take into account the following in order to improve the quality of the manuscript.

Major comments:

1. There are concerns about the absence of details in the methodology adopted and in the lack of clarity of some definitions. It is very important for the authors to confirm if participants with background CKD were excluded or not? It is not clear why the authors did not assess for urinary protein excretion rate (proteinuria)? I suggest that the result of estimation of urinary protein excretion rate should be provided. Also, it is important for the authors to define slow progressors and rapid progressors using the 2012 KDIGO Clinical Practice guideline.

2. Exclusion criteria – How many of this cohort had heart failure and glomerulonephritis? Were these clinical entity excluded?

3. How was acute deterioration of kidney function excluded in the study participants?

4. The sentence “We calculated the intra-twin difference in the annual eGFR change by calculating the eGFR chang in the slow progressor group minus that in the fast progressor group.” (line 122 – 123) should be moved to section on methodology.

5. The authors should state the baseline eGFR for the study cohort.

6. How was hypertension defined in this cohort? I wonder if the rate of CKD progression is related to hypertension, so would be interested in how many slow progressor vs rapid progressor had hypertension. The percentage of slow progressors vs rapid progressors with hypertension should be included in table 1.

7. It is not very clear how the authors arrived at the choice and numbers of independent variables used in the multivariate logistic regression model. I suggest that the regression model should be re-analysed with the inclusion of established risk factors for CKD progression (e.g hypertension and proteinuria). Also the method (backward or forward or stepwise) of inclusion of independent variables in the model should be stated.

8. What were the rates of missing data? If data are complete for all participants, this should be stated.

9. Line 136-138 “in the high discordance group, the slow progressor group showed a higher …………..serum fasting glucose ……..than those of the rapid progressor group” It will be nice to see median (interquartile range) value for serum fasting glucose.

10. Table 1 – The mean±SD values stated for albumin and iron for slow progressor group and rapid progressor group in both low and high discordance twins were exactly the same but the p-values for the 2 pairs were different. I suggest that the authors should double check their data and statistics.

11. Inability to assess the relationship between gut microbiota dysbiosis, nutrient absorption and decline in eGFR should be discussed as part of the limitation.

12. The conclusion did not reflect the exact finding of this study – nutritional related factors were not established as predictors of high discordance of eGFR decline. I suggest a revision to the concluding statement.

Minor comments:

1. No page numbers

2. There are typos that require correction.

Reviewers' comments:

Reviewer's Responses to Questions

**Comments to the Author**

1. Is the manuscript technically sound, and do the data support the conclusions?

Reviewer #1: Partly

Reviewer #2: Partly

2. Has the statistical analysis been performed appropriately and rigorously? 

Reviewer #1: No

Reviewer #2: No

3. Have the authors made all data underlying the findings in their manuscript fully available?

Reviewer #1: Yes

Reviewer #2: Yes

4. Is the manuscript presented in an intelligible fashion and written in standard English?

Reviewer #1: Yes

Reviewer #2: No

5. Review Comments to the Author

Reviewer #1: Thank you for inviting me to review this manuscript “Risk factors associated with the discordance in kidney function decline rate in identical twins.” The authors aimed to analyze the effect of environmental factors on kidney dysfunction in an identical twin cohort.

I have the following concerns with the study.

1. Assessment of kidney function decline takes into account absolute rate of loss of kidney function (Al-Aly Z, Zeringue A, Fu J et al. Rate of kidney function decline associates with mortality. J Am Soc Nephrol 2010; 21: 1961–1969) or percentage change in the kidney function (Cheng TY, Wen SF, Astor BC et al. Mortality risks for all causes and cardiovascular diseases and reduced GFR in a middle-aged working population in Taiwan. Am J Kidney Dis 2008; 52: 1051–1060).

2. The authors categorised participants into high discordant and low discordant and then into rapid progressors and slow progressors in each group based on ≥5% annual eGFR changes. The KDIGO 2012 clinical practice guideline defined progression as a “certain drop” in eGFR defined as a drop in GFR grade associated with a ≥25% decline in eGFR from baseline and “Rapid Progression” as sustained decline in eGFR of more than 5ml/min/1.73m2/yr. I suggest that authors should use clear guidelines in defining technical terms such as rapid progressors.

3. Authors should kindly provide detailed information on the number of times samples were taken to assess renal function during the 3 years period. This is important because studies have shown that the more the frequency of samples collection, the better the confidence and precision in assessing progression of kidney function. Furthermore, do the rapid progressors have a sustained annual eGFR decline of at least 5ml/min/1.73m2/yr

4. Urine albumin excretion estimation such as Urine Albumin-creatinine ratio (ACR) is a recognized predictor of progression of kidney disease. Since this or any other data showing evidence of proteinuria was not included in this study, I suggest it should be included as a limitation.

5. Other predictors of kidney function decline should be included in the multiple logistic regression model with the exception of diabetes since its an exclusion factor. I suggest that in addition to Age and gender, recognized predictors of decline in kidney function must be chosen apriori based on previous literature.

6. I have a strong concern about how the independent variables in the multivariable logistic regression model were derived (i.e difference between slow progressors and rapid progressors). This may have resulted in loss of valuable information during the regression analysis. Also, authors need to demonstrate how they checked for Multicollinearity and other assumptions of Multiple logistic regression model especially for variables like body mass index and body fat percentage. I suggest the authors should use Variance inflation factor. Kindly show the overall statistical details of your Multiple Logistic regression model.

7. Kindly demonstrate evidence of adequacy of sample size.

8. I suggest that Hypoxia inducible factors (HIF) including oxygen-dependent and oxygen-independent HIF-regulators should be discussed in relation to haemoglobin level which is the only independent predictor.

9. Figure 1 should be expanded to provide clarity in the study design. This should show clearly the number of patients in High discordant group and low discordant group. Also,it should clearly show the number of participants in slow progressor and rapid progressor.

Reviewer #2: Major Revision:

Jeong Ah Hwang et al in this study assessed for the risk factors associated with high discordance in annual kidney function decline

in a cohort of identical twins. Though, this is the first study to look at factors associated with discordance in kidney function decline in a cohort of identical twins, this article is very difficult to comprehend. In addition, there are lot of issues with the Materials and method section of this article.

- Authors only excluded diabetic and those who had error in data sets. One is not sure if the cohort they studied have some individuals with background chronic kidney disease. I there are, they need to be excluded from the study. It will add value, if in the result section, the mean(±SD) baseline eGFR of the cohort could be displayed in Table 1.

- The authors did not clearly state what their study outcomes were under the Study outcome section.

- Authors did not clearly define the terms “Rapid progressors and Slow progressors”. Definition of slow/rapid progressors for this study is not clear.

Minor Revision

Abstract,

Background

Line 26:

Line 27 and 28: “We aim to analyze the effect of environmental factors on kidney dysfunction in an identical twin”- This is not the aim of this study. I suggest it is replaced with “This study set out to identify the risk factors for high discordance in kidney function decline in an identical twin cohort.

Methods:

lines 29-31: According to this manuscript, the 333 identical twins were categorized into two groups according to the inter-twin difference in the annual eGFR decline: High discordance twins and low discordance twins, and individual twin were further classified as either a slow progressor or a rapid progressor.

Line 30 : “ eGFR” should be written in full as “estimated glomerular filtration rate (eGFR)”. It can be abbreviated subsequently

Lines Please kindly remove the sentence “identical twins with diabetes were excluded”. This sentence should have come before “The mean difference in the annual …………. who had >5%.........annual eGFR decline.”

Introduction:

Line 1-2:

Lines 48-49 : consider changing the phrase “ but most patients with CKD have an elusive etiology.” to “ however, some patients with CKD have an elusive etiology.”

Line 51

Line 59-61: Please consider changing the phrase “non genetic factors affecting kidney dysfunction besides natural aging in the general population using an identical twin cohort to exclude genetic contribution” to “non genetic factors associated with annual kidney function decline in the general population besides natural aging using an identical twin cohort.”

Materials and Methods:

Study Cohort:

Line 69: Please consider changing “can be found elsewhere “to “have been published elsewhere”

Line 71: twins with background kidney disease should also be excluded from the cohort.

Study Outcome

The study outcomes are not clearly stated in this paragraph. The authors should let us know what their study outcomes are.

Data Collection:

This section should come immediately after Study cohort (Before study outcome.)

Lines 96 and 99 : I suggest the authors replace the word ‘collected’ with “retrieved”

Lines 96/97: Pls kindly remove the sentence “ Body fat percentage was measured by dual X ray absorptiometry”

Statistical Analysis:

Lines 109 and 110: Please remove phrase “to determine whether the mean difference between pairs of measurement was zero or not”.

Lines 111-112: If level of significance was set at 0.05, why were significant variables with (p<0.1) in univariate analysis adjusted for in multivariate analysis ? If level of significance for this study was set at 0.05, please kindly stick to it in your analysis.

Results

Baseline characteristics and annual eGFR change in twins

Line 122 &123: The sentence “We calculated the intra-twin differences in the annual eGFR change in the slow progressor group minus that in the fast progressor group.” Should come under Methods section.

Line 128&129: the subtitle “Difference of clinical characteristics of renal function between the twins in slow or rapid progressor groups- should be changed to “ Baseline Characteristics of the Low and high discordance twins stratified by the rate of annual kidney function decline as either a low progressor or rapid progressor).

Line 130-132: The sentence “To find the factors affecting the intra twin difference in eGFR decline rates, the twins were classified as high discordance and low discordance twins who had >5% and <5% of intra twin difference in annual eGFR change respectively” should have been mentioned under “ study outcome in Materials and Methods”

Line 144: It should be “Risk analysis for high discordance in eGFR decline in twins ”

Line 145-146. The sentences” To find the risk factors associated with high discordance ………….were conducted” and “ Except for age and sex, all baseline characteristics were included as covariates in the form of intra twin differences.” Should be under “S tudy Outcome in Materials and Methods”. These sentences should not be under the “Result section”

Lines 156-160: Fig 2. Blood hemoglobin levels according to the difference in eGFR decline in Identical Twins- should be the title for Fig. 2 while “With dotted line as the center, the plot

on the left ………………low and high groups respectively. Circular dots indicate the slow progressor……. whereas the triangular dots indicate the rapid progressor. The lines connecting the circular and triangular dots represent each pair of twins” should be the legends to Fig 2.

Discussion:

Lines 183-186: The authors stated that they (their study) found that chronic illness and nutritional insufficiency might play a significant role in rapid kidney function decline in the general population.

This study did not demonstrate the above findings. Authors may need to re phrase this sentence.

6. PLOS authors have the option to publish the peer review history of their article (what does this mean?). If published, this will include your full peer review and any attached files.

Reviewer #1: No

Reviewer #2: No

---

## [Author Response · Author response to Decision Letter 0]

28 Sep 2022

Please check the attached file named "Response to reviewers".

---

## [Decision Letter · Decision Letter 1]

8 Nov 2022

PONE-D-22-15736R1Risk factors associated with the discordance in kidney function decline rate in identical twinsPLOS ONE

Dear Dr Ji Eun Kim,

Thank you for submitting your manuscript to PLOS ONE. After careful consideration, we feel that it has merit but but yet to fully meet PLOS ONE’s publication criteria as it currently stands. Therefore, we invite you to submit a revised version of the manuscript that addresses the points raised during the review process. Reviewer #2 has requested for additional clarifications regarding how the twins were stratified in situations when the annual rate of eGFR decline in both twins are the same. Could you kindly provide this information?  Kindly refer to the comments by Reviewer #2 for other typos requiring revision.

We look forward to receiving your revised manuscript.

Kind regards,

Muzamil Olamide Hassan

Academic Editor

PLOS ONE

Journal Requirements:

Reviewers' comments:

Reviewer's Responses to Questions

**Comments to the Author**

1. If the authors have adequately addressed your comments raised in a previous round of review and you feel that this manuscript is now acceptable for publication, you may indicate that here to bypass the “Comments to the Author” section, enter your conflict of interest statement in the “Confidential to Editor” section, and submit your "Accept" recommendation.

Reviewer #1: All comments have been addressed

Reviewer #2: All comments have been addressed

2. Is the manuscript technically sound, and do the data support the conclusions?

Reviewer #1: Yes

Reviewer #2: Yes

3. Has the statistical analysis been performed appropriately and rigorously? 

Reviewer #1: Yes

Reviewer #2: Yes

4. Have the authors made all data underlying the findings in their manuscript fully available?

Reviewer #1: Yes

Reviewer #2: Yes

5. Is the manuscript presented in an intelligible fashion and written in standard English?

Reviewer #1: Yes

Reviewer #2: Yes

6. Review Comments to the Author

Reviewer #1: Thank you for inviting me to review the revised manuscript “Risk factors associated with the discordance in kidney function decline rate in identical twins.”

The authors have made requested corrections in the revised manuscript.

Reviewer #2: Minor Revision:

1. Data Collection: Authors should include how of intra twin difference in annual eGFR decline and other variables (Demographic and clinical variables) were defined for this study.

2. Study group Stratification: Line 95-99. Definition of slow and Rapid progressors and the example given by the authors was noted. However it will be nice for the author to let us know how the twins will be stratified in situations when the annual rate of eGFR decline in both twins are the same. For example, If the rate of eGFR decline in both twins is 1ml/min/1.73m2/yr OR rate of eGFR decline in both twins is 4ml/min/1.73m2/yr. In the 2 scenarios mentioned, how did the authors stratified into rapid / slow progressor when the annual rate of eGFR decline in both twins are the same.

3. Result, Line 156: Please remove the phrase " which was similar to results in the paired t test.

4. Table 1: Please change "Demographic characteristics" to " Demographics and Baseline Clinical Characteristics"

5. Discussion, line 225- 226. Please add reference to the sentence " Although, the exact mechanism...................based on previous studies". There should be a "comma" after the word "Although"

6. Line 267: Better written as " Third, the cause of kidney dysfunction is not known because the cohort only has two time- point measurements (baseline and follow up) and lacks urinalysis.

7. PLOS authors have the option to publish the peer review history of their article (what does this mean?). If published, this will include your full peer review and any attached files.

Reviewer #1: No

Reviewer #2: No

---

## [Author Response · Author response to Decision Letter 1]

12 Dec 2022

Pleased checked the attached word file.

---

## [Decision Letter · Decision Letter 2]

4 Jan 2023

PONE-D-22-15736R2Risk factors associated with the discordance in kidney function decline rate in identical twinsPLOS ONE

Dear Dr. Kim,

Thank you for submitting your manuscript to PLOS ONE. After careful consideration, we feel that it has merit but does not fully meet PLOS ONE’s publication criteria as it currently stands. Therefore, we invite you to submit a revised version of the manuscript that addresses the points raised during the review process.

There is still an outstanding query that the authors are yet to address in the latest revision. The reviewer is asking how will you stratify the twins in situations where the annual rate of eGFR decline in both twins are the same. Taking for example, If the rate of annual eGFR decline in each of the twin is 2ml/min/1.73m2/yr, how will you decide who will be the slow progressor/ rapid progressor group? If none of the participants in the comparison group did not have the same annual eGFR decline, kindly state this clearly in your methodology. Should there be any pair that fit the senario described by the reviewer, kindly provided an explanation how the pair(s) were stratified in the methodology section. Please submit your revised manuscript by Feb 18 2023 11:59PM. If you will need more time than this to complete your revisions, please reply to this message or contact the journal office at plosone@plos.org. Please include the following items when submitting your revised manuscript:A rebuttal letter that responds to each point raised by the academic editor and reviewer(s). You should upload this letter as a separate file labeled 'Response to Reviewers'.A marked-up copy of your manuscript that highlights changes made to the original version. You should upload this as a separate file labeled 'Revised Manuscript with Track Changes'.An unmarked version of your revised paper without tracked changes. You should upload this as a separate file labeled 'Manuscript'.If applicable, we recommend that you deposit your laboratory protocols in protocols.io to enhance the reproducibility of your results. Protocols.io assigns your protocol its own identifier (DOI) so that it can be cited independently in the future. For instructions see: https://journals.plos.org/plosone/s/submission-guidelines#loc-laboratory-protocols. Additionally, PLOS ONE offers an option for publishing peer-reviewed Lab Protocol articles, which describe protocols hosted on protocols.io. Read more information on sharing protocols at https://plos.org/protocols?utm_medium=editorial-email&utm_source=authorletters&utm_campaign=protocols.

We look forward to receiving your revised manuscript.

Kind regards,

Muzamil Olamide Hassan

Academic Editor

PLOS ONE

Journal Requirements:

Reviewers' comments:

Reviewer's Responses to Questions

**Comments to the Author**

1. If the authors have adequately addressed your comments raised in a previous round of review and you feel that this manuscript is now acceptable for publication, you may indicate that here to bypass the “Comments to the Author” section, enter your conflict of interest statement in the “Confidential to Editor” section, and submit your "Accept" recommendation.

Reviewer #2: (No Response)

2. Is the manuscript technically sound, and do the data support the conclusions?

Reviewer #2: Yes

3. Has the statistical analysis been performed appropriately and rigorously? 

Reviewer #2: Yes

4. Have the authors made all data underlying the findings in their manuscript fully available?

Reviewer #2: Yes

5. Is the manuscript presented in an intelligible fashion and written in standard English?

Reviewer #2: Yes

6. Review Comments to the Author

Reviewer #2: Minor:

Authors are yet to answer one of my questions in the last review. The Authors' response to the question was for intra twin difference in eGFR decline and did not address my concern.

1. Study group Stratification: Definition of slow and Rapid progressors and the example given by the authors were noted i.e. when twins have an eGFR decline rate of 1 and 3 mL/min/1.73 m2 98 /yr, then one of the twins with an eGFR decline rate of 3 mL/min/1.73 m2 99 /yr is included in the rapid eGFR change group and the other twin with an eGFR decline rate of 1 mL/min/1.73 m2 100 /yr was included in the slow eGFR group.

My Question: How did the authors stratify the twins in situations where the annual rate of eGFR decline in both twins are the same. For example, If the rate of annual eGFR decline in each of the twin is 2ml/min/1.73m2/yr, how will you decide who will be the slow progressor/ rapid progressor group.

2. lines 92-94 should be written as "The annual eGFR decline rate was calculated using the formula: (eGFR at follow-up - eGFR at baseline)/follow-up duration (year). The intra-twin difference in eGFR decline rate was calculated by subtracting eGFR decline rate of the slow progressor twin from the eGFR decline rate of the other twin (who is the rapid progressor). Intra-twin difference of other baseline clinical characteristics were also calculated using the formula: (value in 'Slow progressor' - value in 'Rapid progressor')

Line 145: Slow progressor " not low progressor"

7. PLOS authors have the option to publish the peer review history of their article (what does this mean?). If published, this will include your full peer review and any attached files.

Reviewer #2: **Yes: **Bolanle Aderonke Omotoso

---

## [Decision Letter · Decision Letter 3]

6 Mar 2023

PONE-D-22-15736R3Risk factors associated with the discordance in kidney function decline rate in identical twinsPLOS ONE

Dear Dr. Kim,

Thank you for submitting your manuscript to PLOS ONE. After careful consideration, we feel that it has merit but does not fully meet PLOS ONE’s publication criteria as it currently stands. Therefore, we invite you to submit a revised version of the manuscript that addresses the points raised during the review process.

Kindly revise the methodology section in line with reviewer's comments highlighted in yellow below.Thank you

We look forward to receiving your revised manuscript.

Kind regards,

Muzamil Olamide Hassan

Academic Editor

PLOS ONE

Journal Requirements:

Reviewers' comments:

Reviewer's Responses to Questions

**Comments to the Author**

1. If the authors have adequately addressed your comments raised in a previous round of review and you feel that this manuscript is now acceptable for publication, you may indicate that here to bypass the “Comments to the Author” section, enter your conflict of interest statement in the “Confidential to Editor” section, and submit your "Accept" recommendation.

Reviewer #2: All comments have been addressed

2. Is the manuscript technically sound, and do the data support the conclusions?

Reviewer #2: Yes

3. Has the statistical analysis been performed appropriately and rigorously? 

Reviewer #2: Yes

4. Have the authors made all data underlying the findings in their manuscript fully available?

Reviewer #2: Yes

5. Is the manuscript presented in an intelligible fashion and written in standard English?

Reviewer #2: Yes

6. Review Comments to the Author

Reviewer #2: Thank you for addressing the concerns on study group classification. However the authors are yet to include this in the method section of this manuscript. Please, I suggest you kindly make amends.

STUDY GROUP CLASSIFICATION Page 6:

1. Line 98-99: I suggest you re write the first sentence as

"All twins were assigned to either the rapid or slow eGFR change groups based on their annual eGFR decline rate"

2. I suggest that the sentence " For pair of twins with the same annual eGFR decline rate, we randomly included each twin in each group, rather than manually selecting those with slow or rapid eGFR progression based on the aforementioned criteria." should come after the sentence "All twins were included in the relatively rapid or slow...............................in slow eGFR change" (lines 100-104)

3. Delete the sentence "That is, even if both have a high eGFR decline rate (e.g. 7 ml/min/1.73m2/yr for both twins) or both have a low eGFR decline rate (e.g. 1 ml/min/1.73m2/yr for both twins), they were classified as the same group if the intra-twin difference of eGFR decline rate in each pair was the same. (lines 1111-113)

7. PLOS authors have the option to publish the peer review history of their article (what does this mean?). If published, this will include your full peer review and any attached files.

Reviewer #2: **Yes: **Bolanle Aderonke Omotoso

---

## [Editor Report · Decision Letter 4]

27 Mar 2023

Risk factors associated with the discordance in kidney function decline rate in identical twins

PONE-D-22-15736R4

Dear Dr. Kim,

We’re pleased to inform you that your manuscript has been judged scientifically suitable for publication and will be formally accepted for publication once it meets all outstanding technical requirements.

Kind regards,

Muzamil Olamide Hassan

Academic Editor

PLOS ONE
---

## [Editor Report · Acceptance letter]

4 Apr 2023

PONE-D-22-15736R4 

Risk factors associated with the discordance in kidney function decline rate in identical twins 

Dear Dr. Kim:

I'm pleased to inform you that your manuscript has been deemed suitable for publication in PLOS ONE. Congratulations! Your manuscript is now with our production department. 

Kind regards, 

on behalf of

Dr. Muzamil Olamide Hassan 

Academic Editor

PLOS ONE